# Optimization of Oil Extraction from Rice Bran with Mixed Solvent Using Response Surface Methodology

**DOI:** 10.3390/foods11233849

**Published:** 2022-11-28

**Authors:** Zhenhua Wang, Shuzhen Li, Min Zhang, Huanyue Yang, Gang Li, Xin Ren, Shan Liang

**Affiliations:** 1Beijing Advanced Innovation Center for Food Nutrition and Human Health, Beijing Technology and Business University, Beijing 100048, China; 2Beijing Engineering and Technology Research Center of Food Additives, Beijing Technology and Business University, Beijing 100048, China

**Keywords:** mixed solvent, rice bran, extraction, response surface methodology

## Abstract

In order to improve the extraction ratio of rice bran oil, a single-factor experiment and response surface methodology with a central composite design were used to determine a new mixed solvent and the optimal extraction conditions of the mixed solvent. The effects of solid–liquid ratio, extraction time, extraction temperature, and oscillation speed on the extraction ratio were investigated. The regression equation was established, and the optimal extraction conditions were determined as follows: a solid–liquid ratio of 5.5:1, extraction temperature of 45 °C, extraction time of 12 min, and extraction ratio of rice bran oil of 85.8%. Compared with traditional solvent extraction, the peroxide value, acid value, iodine value, and fatty acid composition content of rice bran oil extracted using the new mixed solvent were close to those of *n*-hexane and significantly lower than those of solvent No. 6, while the content of oryzanol and total sterol increased to 2.7% and 5.1%. This study can be useful in exploring the possibility of new mixed solvents and provide theoretical guidance and data support for the production practice of new mixed solvents.

## 1. Introduction

As one of the most important cereal crops, rice accounts for about 22.3% of global cereal grain production with an annual planting area of about 736 million hectares in 2020 [1], and it is one of the staple foods for more than half the world’s population [2]. Rice bran is a valuable byproduct of rice production, which accounts for approximately 8% of the rice grain and 15–20% of the oil [3]. Rice bran oil (RBO) is reported to have nearly all types of nutrients among vegetable oils. As a deep-processing product of rice bran, RBO is considered to be rich in a variety of nutritional components such as oryzanol, tocopherol, and tocotrienol [4], as well as phytosterols and squalene [5], and it is attractive for its unique nutraceutical properties and balanced fatty acid composition [6]. Among them, oryzanol is an important component unique to RBO, and it has been shown to have biological and physiological functions such as lowering serum cholesterol, anticarcinogenic antioxidant properties, and reducing allergic inflammation [7].

The solvent extraction method is commonly used by large oil companies because of its high oil yield, low cost, and easy realization of large-scale automated production. The most commonly employed solvent is a mixture of isomers of hexane, derived from petroleum distillation in the conventional oil extraction process [8], which is flammable and explosive, along with high energy consumption and air pollution, and it was found that human exposure to *n*-hexane vapor for a long time can cause damage to the central nervous system and motor nerve cells [9]. Moreover, aromatic hydrocarbons, halogenated hydrocarbons, furfural, and other solvents are highly toxic and harmful to health; diethyl ether and isopropyl ether are easily oxidized and have poor stability; *n*-heptane has a high boiling point and large energy consumption for recycling; petroleum ether is volatile, resulting in easy solvent loss; propane and butane in hydrocarbons have low boiling points and poor operating safety. Most solvents are harmful to health, more expensive, and not suitable for industrial production. Therefore, finding a safe, green, and new extraction solvent enabling high oil yield is of great significance to reduce the harm caused by *n*-hexane and improve product recovery ratio and quality.

At present, studies on the replacement of *n*-hexane solvents have mainly focused on two solvents: short-chain alcohols and short-side-chain hydrocarbon compounds other than *n*-hexane. However, each solvent has its own advantages and disadvantages. For example, isopropanol can be used for high-quality oil extraction and improving the sensory and functional properties of the defatted meal due to the higher operational safety, bio-renewability, and low toxicity [10]. Cyclohexane is noncorrosive with low toxicity and no carcinogenic mutagenic effects, and its price is equivalent to *n*-hexane. Absolute ethanol is a renewable solvent, and its operation is relatively safe. In addition, *n*-pentane is a short-chain hydrocarbon compound, which has a similar carbon chain, as well as similar physical and chemical properties, to *n*-hexane but n-pentane has low toxicity and a low boiling point. The damage to the human body is minimal, and the extraction temperature is low, which can retain effective ingredients in the meal and improve its comprehensive utilization value. Therefore, using mixed solvents has become a trend in the study of new extraction solvents [11].

In the present study, the influences of solvent type (isopropanol, cyclohexane, absolute ethanol, *n*-hexane, and *n*-pentane) on RBO yield were evaluated, and a novel mixed solvent was proposed. The most significant variables affecting the oil extraction process were selected through a single-factor experimental design, and then the response surface methodology (RSM) was used to optimize extraction conditions using a central composite design (CCD), with RBO yield as the indicator. A second-order polynomial equation was used to represent RBO yield as a function of the solid–liquid ratio, extraction time, and extraction temperature. Furthermore, the novel mixed solvent was compared with the single solvents in terms of RBO yield and oryzanol content.

This study aimed to investigate the effects of solid–liquid ratio, extraction time, extraction temperature, and oscillation speed on the new mixed solvent extraction ratio and to find the optimum operating conditions to maximize the extraction ratio.

## 2. Materials and Methods

### 2.1. Materials

Freshly extruded rice bran was provided by Heilongjiang Beidahuang Rice Industry Group Co., Ltd. (Harbin, China) and then quickly milled using a grinder (A11, IKA, Staufen, Germany) under the protection of liquid nitrogen. The milled powder was sieved through 20 mesh and stored at −20 °C to prevent enzymatic deterioration before the experiments. The largest rice bran particle size was about 1400 μm, and the average diameter was 213.6 μm.

The moisture content of the rice bran was 8.60% ± 0.05% (*w*/*w*), which was determined using the oven drying method at 105 ± 2 °C. The oil content and protein content of the rice bran were 20.64% ± 0.02% and 14.47% ± 0.02%, determined according to the national standard methods GB 5009.6-2016 and GB 5009.5-2016, respectively.

Isopropanol (boiling point: 82.40 °C), cyclohexane (boiling point: 80.72 °C), *n*-hexane (boiling point: 68.70 °C), *n*-pentane (boiling point: 36.10 °C), petroleum ether (boiling point range: 30–60 °C), *n*-heptane (boiling point: 98.50 °C), and absolute ethanol (boiling point: 78.32 °C) were purchased from Fuchen (Tianjin, China) Chemical Reagent Co., Ltd. in China. All solvents in the extraction process were analytical grade.

No. 6 solvent (*n*-hexane, 2,4-dimethylpentane, 2,3-dimethylbutane, cyclopentane, cyclohexane, benzene, and n-pentane accounting 30%, 18%, 18%, 10%, 8%, 4%, and 2%, respectively, along with some low-content components such as 3-methylpentane and 2,2,3-trimethylbutane, which are complex in composition) was purchased from Beijing Chemical Works Co., Ltd. (Beijing, China) in China.

### 2.2. Oil Extraction and Solvent Screening

First, 5 g of rice bran was put into a conical flask and then extracted using different solvents with a certain solid–liquid ratio at different temperatures in a water bath oscillator (THZ-83A, Jiangxing, Shanghai, China). After extraction, the solvent–oil mixture was filtered in a flat-bottomed flask, and the solvents were evaporated under a vacuum using a rotary evaporator (RE-2000A, Yarong, Shanghai, China) at 45 °C to obtain the oil. Subsequently, the oil was further dried in a fume cupboard (WG9220A, Tongli Xinda, Tianjin, China), and then weighed using an electronic analytical balance (FB204, Yoke, Shanghai, China). The RBO yield was calculated gravimetrically using Equation (1).
(1)$$M=\left(m_{2}-m_{1}\right)/\;\left(m_{0}\times M_{0}\right),$$
where M is the RBO yield (%), m_1_ is the mass of the empty flat-bottomed flask (g), m_2_ is the mass of the flat-bottomed flask with oil after the rotary evaporation and drying process (g), m_0_ is the mass of rice bran (g), and M_0_ is total oil content in the raw material (%).

Under the conditions of solid–liquid ratio = 5:1, extraction temperature = 50 °C, extraction time = 30 min, and oscillation speed = 180 r/min, five solvents (isopropanol, absolute ethanol, cyclohexane, *n*-hexane, and *n*-pentane) were used for extraction, and RBO yield and its oryzanol content were measured. Two solvents with better extraction ability were selected and mixed at ratios of 1:9, 2:8, 3:7, 4:6, 5:5, 6:4, 7:3, 8:2, and 9:1. The optimal mixing ratio of the mixed solvent was determined by investigating the RBO yield and the oryzanol content extraction effects of different compounding solvents.

### 2.3. Single Factor Experiment

The main factors that influenced the extraction effect were the solid–liquid ratio, extraction time, extraction temperature, and oscillation speed. Each variable consisted of five levels: solid–liquid ratio (2:1, 3:1, 4:1, 5:1, and 6:1, *v*/*w*); extraction temperature (20, 30, 40, 50, and 60 °C); extraction time (5, 10, 20, 35, and 55 min); oscillation speed (90, 120, 150, 180, and 210 r/min). When one variable changed, the other three variables remained the same.

### 2.4. Determination of RBO Properties

The acid value, iodine value, peroxide value, moisture and volatile content, and color of the RBO were determined according to previous studies with some modifications [7,12].

#### 2.4.1. Physicochemical Indexes

Peroxide value was determined according to GB5009.227-2016. Iodine value was determined according to GB/T 5532-2008. The acid value was determined according to GB5009.229-2016. The color was determined by CR-400/410 chromaticity meter. Moisture content and volatile content were determined according to GB5009.236-2016.

#### 2.4.2. Fatty Acid Composition and Content

The fatty acid composition of RBO was assessed as fatty acid methyl esters and determined by GC-MS (QP2010, Shimadzu, Tokyo, Japan). The RBO (0.1 g) were weighed by electronic balance (FB204, Youke, Shanghai, China) and mixed with 8 mL NaOH-methanol solution (2:98, *v*/*v*), watered bath by Electric thermostatic water bath (DK-98-A, Taist, Tianjin, China) at 80 °C for 40 min, then 7 mL 15% boron trifluoride methanol solution was added and water-bath at 80 °C for 2 min and cooled to room temperature. Moderate sodium chloride and 10 mL n-heptane were added, shaking extraction was performed for 10 min, and then the supernatant was added with moderate anhydrous sodium sulfate for dewatering and diluting 10 times.

Chromatographic conditions: TR-WAXMS capillary column (Thermo Scientific, Waltham, MA, USA, 30 m × 0.25 mm × 0.25 μm), injection temperature: 25 °C. Heating procedure: the initial column temperature was maintained at 80 °C for 2 min, then the column temperature increased to 210 °C at 15 °C/min and to 240 °C at 2 °C/min. The carrier gas was ultra-pure helium gas with a flow rate of 1 mL/min and split ratio of 10:1.

MS-conditions: electron–ion source, electron energy of 70 eV, ion source temperature of 200 °C, interface temperature of 240 °C, mass scanning range *m*/*z*: 45-500. Pristine data were retrieved online by NIST 11 Library, and identification results with matching degrees and purity greater than 800 (maximum 1000) were selected. The retention index of fatty acid methyl esters in RBO was compared with that of the mixed standard of fatty acid methyl esters, and the relative percentage of each component was calculated by a peak area normalization method.

#### 2.4.3. Oryzanol Content

The oryzanol content in RBO was determined according to the method of China Grain Sector Standard LS/T 6121.1-2017 using a UV-VIS spectrophotometer (UV2800-A, UNICO, Shanghai, China) at 315 nm. Briefly, RBO samples were dissolved in heptane. The oryzanol content was calculated by Equation (2):X = (A × V × N)/ (m × 359)(2)
where X represents the oryzanol content in RBO (%), A is the absorbance of samples, V is the constant volume of the solution to be tested (mL), N is the dilution factor after samples volume determination (N = 1 if not diluted), m is the mass of samples (g), 359 is known as the specific extinction coefficient of oryzanol (g/100 mL).

#### 2.4.4. Total Sterol Content

100 mg RBO was weighed and mixed with 5 mL potassium hydroxide-ethanol solution (1 mol/L), then watered bath at 80 °C for 1 h and cooled to room temperature, then mixed with 10 mL petroleum ether and shaken 5 min. After stratification, 1 mL supernatant was dried by Termovap Sample Concentrator (NDK200-1, MIU, Shanghai, China) and diluted with anhydrous ethanol in a 10-mL volumetric flask and mixed. Then, 2 mL sample solution was mixed with 2 mL anhydrous ethanol and 2 mL phosphofer-sulfur chromogenic agent, and determined by a UV-VIS spectrophotometer (UV2800-A, UNICO, Shanghai, China) at 520 nm. According to the standard curve, the total sterol content in RBO was calculated by Equation (3):G = (m × V × v_1_)/ (M × v_2_)(3)
where G is the total sterol content (%); m is the weight of stigmasterol in the standard curve corresponding (μg); V is a constant volume (mL); v_1_ is the volume of petroleum ether (mL); M is the mass of rice oil (g); v_2_ is the volume of petroleum ether (mL).

### 2.5. The RSM Experimental Design

According to the results of single-factor experiments, three major influence factors were chosen for RSM analysis. Then, CCD was used to determine the optimum process conditions for the extraction of RBO. Solid–liquid ratio (A), extraction time (B), and extraction temperature (C) were chosen as independent variables, and the yield of RBO (R) was taken as the response value of the design experiment. Through CCD, 20 experiments with different combinations of the three factors were carried out, and the results are shown in Table 1.

According to the CCD configuration, the relationship between the independent variables and the response values is shown in Equation (4).
(4)$$R=k_{0}+k_{1}A+k_{2}B+k_{3}C+k_{12}\mathrm{AB}+k_{13}\mathrm{AC}+k_{23}\mathrm{BC}+k_{123}\mathrm{ABC}+k_{11}A^{2}+k_{22}B^{2}+k_{33}C^{2},$$
where R is the predicted response value, A_0_ is the regression intercept, k_i_, k_ij_, and k_ijk_ (i, j, k = 1, 2, 3) are the linear, quadratic, and interactive regression coefficients, and A, B, and C are coded independent variables.

### 2.6. Statistical Analysis

All experimental procedures were performed three times, and data were expressed as means ± standard deviations (SDs), which were analyzed statistically. One-way analysis of variance was used to estimate differences among group average values with the application of Duncan’s tests using SPSS 17. Design Expert Version 8.0.6 was used for regression analysis, response surface plot, and optimization.

## 3. Results

### 3.1. Single Factor Optimization Results of Extraction Process

#### 3.1.1. Effect of Solvent Type and Mixing Ratio on RBO Yield

The solvent type was important to the RBO yield, as shown in Figure 1. In terms of RBO yield, isopropanol, cyclohexane, and *n*-hexane all exhibited better performance, reaching 82.6%, 81.9%, and 81.5%, respectively, followed by *n*-pentane and absolute ethanol. With regard to the oryzanol content of RBO, it reached a maximum of 2.4% when using isopropanol and cyclohexane as the extraction solvents. The extraction effect was better with no significant difference, while that of *n*-hexane was worse, reaching 2.3% (*p* < 0.05). Nonpolar solvents can reduce the extraction of polar substances such as polysaccharides and improve extraction efficiency [13]. Moreover, Comerlatto et al. [14] used two solvents, isopropanol and ethanol, for soybean oil extraction. The study showed that isopropanol could be used as a substitute solvent for *n*-hexane, whereby the oil close to the particle surface was quickly removed using pure isopropanol, thus indicating that extraction was promoted. Moreover, Seth et al. [15] found that isopropanol resulted in higher extraction rates and oil recoveries than *n*-hexane, while similar results were obtained by Fraterrigo et al. [16]. Adding a nonpolar solvent cyclohexane to isopropanol as an extraction solvent not only meets the requirements of no great change to the existing extraction equipment, but also improves the extraction efficiency, which can be easily applied in the oil industry. Additionally, cyclohexane is noncorrosive with low toxicity and no carcinogenic mutagenic effect, and its price is equivalent to that of *n*-hexane. Hence, isopropanol and cyclohexane were selected for further investigation of their impacts on RBO yield.

Figure 2 shows the RBO yield extraction using different mixing ratios of mixed solvent. It was observed that the combination of isopropanol and cyclohexane (1:1, *v*/*v*) was advantageous for oil extraction and was significantly higher compared to the single solvents, reaching 85.0%; the oryzanol content of RBO reached 2.6% (*p* < 0.05). The positive effect of the solvent mixture on the RBO yield could be attributed to the combination of polar and nonpolar solvents and the combination of the advantages of alcohols and alkanes. The addition of appropriate amounts of polar solvents, isopropanol and ethanol, can promote extraction [14]. Isopropanol is a polar solvent, which can effectively destroy the binding force among polar proteins, such as membrane proteins [6]. The liposome membrane is loose and porous, and nonpolar solvent cyclohexane could penetrate into the liposome, when well combined with the RBO. Furthermore, acetone mixed with methanol (1:1, *v*/*v*) was selected by Sivagnanam et al. [17] for oil extraction, exhibiting the same effect as *n*-hexane. Hence, a solvent mixture of isopropanol and cyclohexane (1:1, *v*/*v*) was used in further experiments.

#### 3.1.2. Effect of Solid–Liquid Ratio on RBO Yield

The solid–liquid ratio is one of the important factors influencing the RBO yield. In general, when the solid–liquid ratio reaches a certain level, the oil yield is basically balanced. The effect of solid–liquid ratio on RBO yield is shown in Figure 3A. When the solid–liquid ratio increased, the concentration gradient of oil between the rice bran surface and the solvent increased, thereby resulting in a high oil extraction yield [18], and different solid–liquid ratios (6:1, 8:1, and 10:1 *w*/*v*) were selected by Shet et al., obtaining similar results. However, excessively increasing the solid–liquid ratio did not significantly increase the oil yield, as presented by Zhang et al. [19] and Wang et al. [20]. Go et al. [21] changed the solid–liquid ratio from 1:1 to 2.5:1 and from 4:1 to 12:1; however, the RBO yield did not change significantly. Additionally, more solvent increases the production costs and difficulty of subsequent solvent recovery. It is interesting that Juchen et al. [22] found that the initial extraction rate could be maximized for a specific solid–liquid ratio. Beyond this amount, excess solvent at the extraction began to limit the mass transfer of oil to the solvent phase. Hence, it is wise to use the optimal solid–liquid ratio at the beginning of each extraction in a sequential process, representing a suitable approach to achieving high oil yields with reduced solvent expenditure and extraction periods, which are essential parameters for a feasible and optimized industrial extraction process. Therefore, the solid–liquid ratio was selected as 5:1 to investigate the effect of other variables on RBO extraction.

#### 3.1.3. Effect of Extraction Time on RBO Yield

Figure 3B shows the effect of extraction time on RBO yield. The figure shows a rapid extraction in the initial period followed by a slower extraction rate, with 82.6% of the oil being extracted in the first 10 min (*p* < 0.05). A study by Wang et al. [20] showed that the RBO yield was 17.5% at 1 h, but there was no significant difference from 1 to 2.5 h, indicating that the majority of RBO was extracted in a short time. Furthermore, Go et al. [21] found that, regardless of conditions, 90% of the extractable lipids were solubilized in hexane in less than 10 min. Similarly, the study of Azevedo et al. [23] also revealed that the oil extraction was sufficiently completed in 30 min, with the full extraction taking 0–90 min. Because oil extraction is essentially a process of dynamic mass transfer [11,24], a short extraction time makes it difficult for the solvent to fully come into contact with the oil and form a mass transfer balance, thus resulting in incomplete extraction and a low oil yield [25]. When the mass transfer balance is reached, the oil yield is not improved obviously with time [12]. Therefore, the driving force is reduced after the initial period, and the extraction rate is lowered [26]. The RBO yield at 55 min was 83.5%, which was only 0.01 times higher than that at 10 min. Other substances can easily be extracted over such a long time, which increases the specific energy consumption and production cost. Hence, 10 min was selected as a suitable time for RBO extraction.

#### 3.1.4. Effect of Extraction Temperature on RBO Yield

It can be seen from Figure 3C that the RBO yield increased with temperature, as also shown by Kamimura et al. [18] and Toda et al. [27]. The molecular kinetic energy and the disordered thermal motion increased with temperature, while the diffusion resistance and the oil viscosity decreased; moreover, the solubility of RBO increased with temperature, which was beneficial to oil extraction [28]. The RBO yield increased sharply when the temperature increased from 20 °C to 40 °C (*p* < 0.05), but it was little affected by the temperature upon reaching 40 °C (*p* < 0.05). Similarly, the result of extraction by Benito-Román et al. [29] was analogous to the extraction temperature changing from 40 to 60 °C. Some studies by Pinto et al. [30] and Herawati et al. [31] showed that a high temperature may affect the fatty acid composition and destroy some heat-sensitive substances in rice bran oil, indicating that excessive temperature is not always suitable for the extraction process and may cause undesirable effects. In addition, a high temperature led to high energy consumption, the destruction of the active nutrients in the oil, and a deepening of the oil color [32]. In industrial production, a low temperature is generally applied for oil extraction in order to minimize energy consumption, reduce production costs, and avoid oxidization of the oil [33]. Hence, 40 °C was selected as a suitable temperature for RBO extraction.

#### 3.1.5. Effect of Oscillation Speed on RBO Yield

During the extraction process, the solvent penetrates into the rice bran granules and combines with the oil, which diffuses from the rice bran granules into the solvent [7]. The oscillating extraction method can promote the diffusion of oil from the surface of the raw material into the solvent as quickly as possible to reach a uniform state. Figure 3D shows that oscillation speed had no significant effect on RBO extraction (*p* < 0.05). When the oscillation speed was 90 r/min, 120 r/min, 150 r/min, 180 r/min, and 210 r/min, the RBO yield reached 81.9%, 83.6%, 83.9%, 84.5%, and 83.9%, respectively. This shows that an excessively high oscillation speed resulted in a lower RBO yield and higher energy consumption and production costs. It was proven that an appropriate oscillation speed was advantageous for the solvent and the oil to uniformly and fully come into contact and diffuse. Therefore, 180 r/min was selected as the oscillation speed for oil extraction, but it was not selected for the response surface test.

### 3.2. Optimization of Extraction by RSM

#### 3.2.1. The RSM Results of the Extraction Process

The relationship between variables and response values were established on the basis of the experimental results of CCD and regression analysis (Table 1).

Design Expert Version 8.06 was used to perform quadratic multiple regression fitting on the results in Table 1 to obtain the following quadratic polynomial equation:(5)$$R=83.87+2.89A+1.04B+2.08C-0.52\mathrm{AB}+0.054\mathrm{AC}-0.55\mathrm{BC}-1.72A^{2}-0.51B^{2}-1.11C^{2},$$
where R is the RBO yield, and A, B, and C are the coded independent variables.

Table 2 shows the variance analysis of the above regression model. The significance of the coefficients of each variable in the regression equation was checked using the F-test. The factors, A, B, C, A^2^, C^2^ (*p* < 0.01), BC, and B^2^ (*p* < 0.05) had statistically significant effects on the RBO yield, while the effects of other interactions were secondary. The model F-value for RBO yield was 57.86 (*p* < 0.0001) indicating that the regression model reached a significant level. The lack of fit of the F-value was 3.06 (*p* > 0.05) for RBO yield, i.e., not significant, indicating that the stability of this model was good. The coefficient of determination R^2^ = 0.981, with the correction coefficient R^2^_Adj_ = 0.964, indicated that the response model had high relativity, good fitting, and a small experimental error; thus, it was suitable to simulate the correlation among various factors and response values from the regression model. Comparing the F-values of the three factors showed that the effects on the extraction rate of rice oil were A > C > B in order. The quadratic coefficients in the model were all negative, indicating that the paraboloid of the model was downward, and there was an extremely optimal extraction point of rice oil, indicating that the regression equation had a good fit, and that the results were reliable.

#### 3.2.2. Optimization of the Extraction Conditions and Verification of the Model

The 3D analysis diagram of the response surface and its contour lines are shown in Figure 4. The RBO yield increased first and then tended to be gentle with the increase in solid–liquid ratio, temperature, and time, indicating that the model had a maximum value. When the temperature was kept at the central level (40 °C), the response surface curves of the solid–liquid ratio and time were steeper, indicating that they had a significant effect on the RBO yield, which is similar to the study by Wang et al. [20] (Figure 4a). The contour map of the interaction of solid–liquid ratio and time was round, indicating that their interaction was not significant, in contrast to the report by Jia et al. [34], which might be due to different cultivars of rice bran and extraction methods. According to the determination of extraction time at the center level (10 min), the steepness of the response surface fluctuated greatly with the changes in solid–liquid ratio and temperature, indicating that the solid–liquid ratio and temperature had a significant effect on the RBO yield (Figure 4b). The contour map of the solid–liquid ratio and temperature interaction was round, indicating that their interaction was not significant. When the solid–liquid ratio was kept at the center level (5:1), the response surface curves of time and temperature were steep, indicating that the time and temperature had a significant effect on the RBO yield (Figure 4c). The contour map of the interaction of time and temperature was elliptical (*p* < 0.05), indicating that their interaction had a significant effect.

According to the mathematical model obtained from the RSM, the optimal process variables were a solid–liquid ratio of 5.52:1, the extraction temperature of 44.9 °C, and extraction time of 11.31 min; the predicted RBO yield was 85.8%. Considering the limitations of the actual extraction operation, the optimized conditions were determined as a solid–liquid ratio of 5.5:1, extraction temperature of 45 °C, and the extraction time of 12 min; the RBO yield was 85.8%. The measured value and the predicted value were the same. Therefore, the fitted function model was credible, and the response surface methodology was practical.

According to the optimal extraction conditions, a comparative analysis of RBO yield and oryzanol content was conducted using the mixed solvent and the single solvents isopropanol and cyclohexane. The results showed that the RBO yield extracted using the mixed solvent reached 85.8% ± 0.0%, and the oryzanol content reached 2.5% ± 0.0% (Table 3), which was significantly larger than the results using the single solvents under optimal process conditions. Thus, it was proven to be feasible and superior to extract RBO using the mixed solvent rather than isopropanol and cyclohexane.

### 3.3. Effect of Mixed Solvent on the Properties of RBO

The physicochemical properties of RBO extracted using different solvents are shown in Table 4. The moisture and volatile content in RBO extracted using the mixed solvent was significantly lower than that using solvent No. 6, but higher than that using *n*-hexane (*p* < 0.05). The rancidity and oxidation of RBO are indicated by the acid value and peroxide value, respectively. Oil samples with a low acid value and peroxide value are considered of high quality. In this study, the peroxide value, acid value, and iodine value using the mixed solvent method were close to those using *n*-hexane and significantly lower than those using solvent No. 6 (*p* < 0.05), which was beneficial to improving the oil refining rate and product quality. The acid and peroxide values were slightly different from those reported by Phan et al. [35], which might be due to the climatic conditions and different varieties of rice bran. In contrast, the acid and peroxide values were found to be higher than those reported by Xu et al. [36], which could be due to the lower free fatty acid and peroxide of rice bran oil extracted using aqueous enzymatic extraction. Furthermore, the iodine value of the mixed solvent was found to be higher than that of the No. 6 solvent, which could be due to a higher content of unsaturated fatty acids in No. 6 solvent extraction.

Compared with *n*-hexane and solvent No. 6, the values of L* and b* were lower and the value of a* was higher, indicating that the color of RBO was deeper. Excluding the influence of raw materials and process conditions, a possible reason is that isopropanol in the new mixed solvent is a polar solvent, which can extract more polar substances such as protein and sugar. These polar substances have a mutual complexation reaction, resulting in the deepening of oil color [37]. It is also possible that pigments such as carotene and chlorophyll were extracted from the bran due to organic solvents, as proven by Hu et al. [37].

Oryzanol is the most important antioxidant and bioactive component in rice bran [29]. Sterols are physiologically important active components of vegetable oils. The contents of oryzanol and total sterol using the mixed solvent method were up to 2.7% and 5.1%, i.e., significantly higher than those using *n*-hexane and solvent No. 6 solvent, which could be due to isopropanol being superior for oryzanol and sterol retention [16].

Fourteen major fatty acid components including eight saturated fatty acids, four monounsaturated fatty acids, and two polyunsaturated fatty acids were detected and identified (Table 5). It could be observed that the major fatty acids of RBO were linoleic acid (38.7% ± 0.1%), oleic acid (33.7% ± 0.0%), and palmitic acid (19.9% ± 0.3%), in line with the results of Xu et al. [36]. However, the content and variety of fatty acids were different from the results of Mingyai et al. [38] and Liu et al. [39], which could be attributed to the cultivars, planting regions, climate conditions, and the extraction process. Moreover, there were significant differences in the composition of fatty acids extracted using the three solvents; however, there were no significant differences (*p* < 0.05) in the content of main fatty acids. Therefore, compared with the traditional solvent, the new mixed solvent has considerable extraction ability and active substance conservation superiority.

## 4. Conclusions

RSM with a CCD was used to optimize the extraction conditions of RBO. The optimal conditions were a solid–liquid ratio of 5.5:1, extraction temperature of 45 °C, and extraction time of 12 min; the RBO yield was 85.8%. Compared with the traditional solvent, the new mixed solvent had a similar extraction ability and active substance conservation superiority in terms of physicochemical properties and fatty acid composition and content of RBO. This research can be useful in exploring the possibility of new mixed solvents and provides theoretical guidance and data support for the production of new mixed solvents.

## Figures and Tables

**Figure 1 foods-11-03849-f001:**
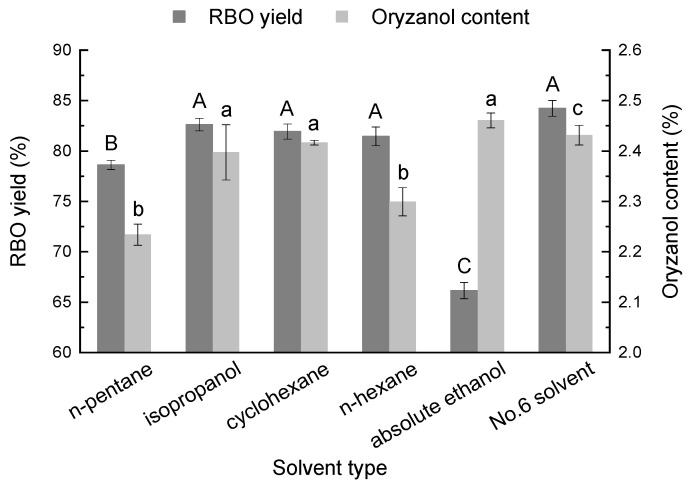
Effect of solvent type on RBO yield. Different uppercase and lowercase above the columns indicate significant differences in RBO yield and oryzanol content (*p* < 0.05).

**Figure 2 foods-11-03849-f002:**
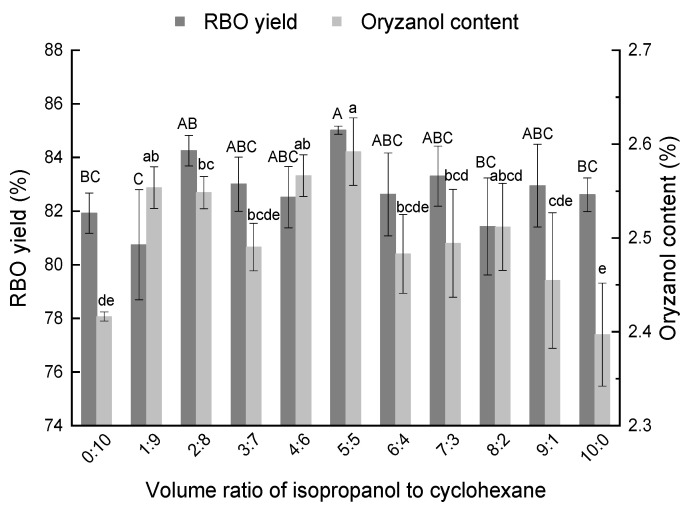
Effect of mixing ratio on RBO yield. Different uppercase and lowercase above the columns indicate significant differences in RBO yield and oryzanol content (*p* < 0.05).

**Figure 3 foods-11-03849-f003:**
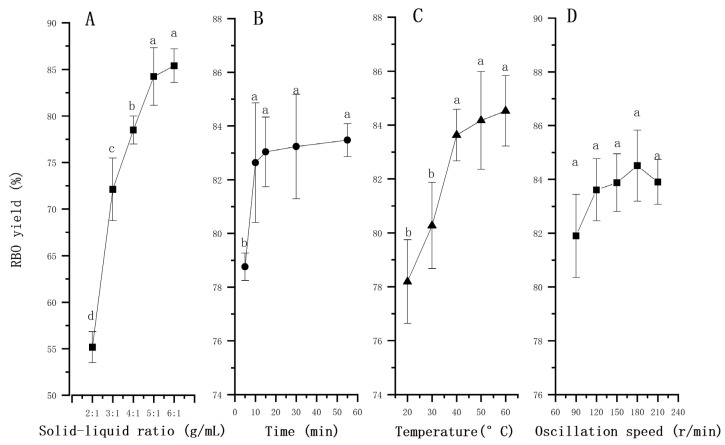
Effects of solid–liquid ratio (**A**), extraction time (**B**), extraction temperature (**C**), and oscillation speed (**D**) on RBO yield. Different lowercase indicate significant differences among groups (*p* < 0.05).

**Figure 4 foods-11-03849-f004:**
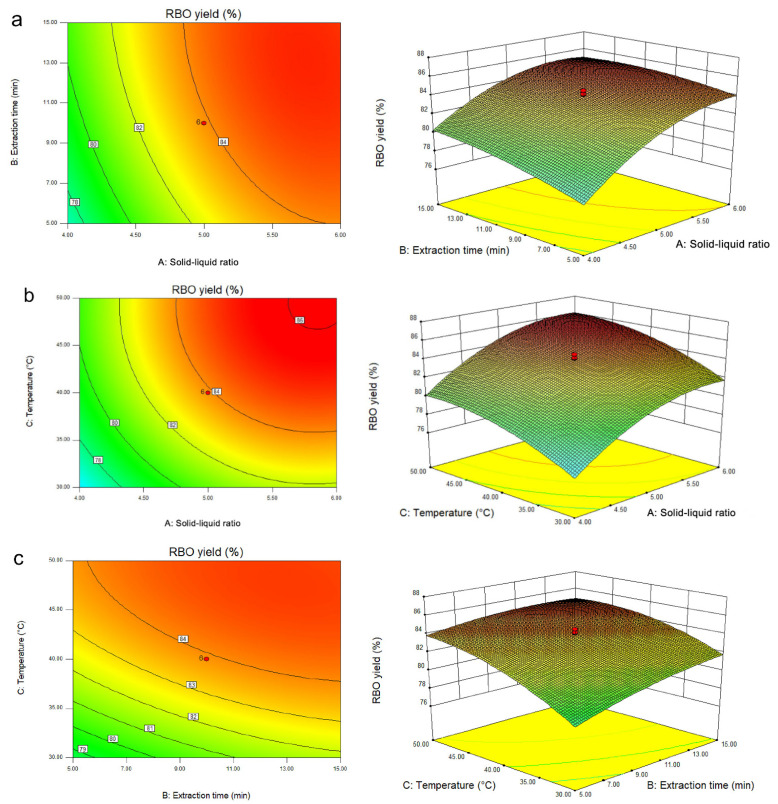
Contour plots and response surface plots of the interactive effects of different factors on RBO yield: (**a**) extraction time and solid–liquid ratio; (**b**) temperature and solid–liquid ratio; (**c**) temperature and time; (A) solid–liquid ratio; (B) extraction time; (C) temperature.

**Table 1 foods-11-03849-t001:** Three-level central composite design with three independent variables and experimental results (yield of RBO).

Run	Solid–Liquid Ratio (A)		Extraction Time (min) (B)		Temperature (°C) (C)		RBO Yield (%)
	Coded Value	Actual Value	Coded Value	Actual Value	Coded Value	Actual Value	
1	0	5:1	1.68	18.41	0	40	83.9
2	0	5:1	0	10	0	40	84.2
3	−1	4:1	−1	5	1	50	79.4
4	−1.68	3.32:1	0	10	0	40	73.2
5	−1	4:1	−1	5	−1	30	73.8
6	0	5:1	0	10	0	40	84.1
7	1	6:1	1	15	−1	30	82.2
8	1	6:1	−1	5	1	50	85.2
9	0	5:1	0	10	−1.68	23.18	76.5
10	0	5:1	0	10	0	40	83.5
11	−1	4:1	1	15	−1	30	78.7
12	1	6:1	−1	5	−1	30	80.7
13	0	5:1	−1.68	1.59	1	40	80.3
14	0	5:1	0	10	1.68	56.82	84.2
15	−1	4:1	1	15	1	50	80.7
16	0	5:1	0	10	0	40	84.6
17	0	5:1	0	10	0	40	83.2
18	1.68	6.68:1	0	10	0	40	84.0
19	0	5:1	0	10	0	40	83.7
20	1	6:1	1	15	1	50	85.7

**Table 2 foods-11-03849-t002:** Analysis of variance and reliability of regression model.

Source	Sum of Squares	Degree of Freedom	Mean Square	F-Value	*p*-Value	
Model	248.58	9	27.62	57.86	<0.0001	significant
A	113.73	1	113.73	238.24	<0.0001	
B	14.64	1	14.64	30.66	0.0001	
C	59.08	1	59.08	123.77	<0.0001	
AB	2.17	1	2.17	4.55	0.0586	
AC	0.023	1	0.023	0.048	0.8303	
BC	2.45	1	2.45	5.14	0.0468	
A^2^	42.69	1	42.69	89.42	<0.0001	
B^2^	3.70	1	3.70	7.75	0.0193	
C^2^	17.80	1	17.80	37.28	0.0001	
Residual	4.77	10	0.48			
Lack of fit	3.60	5	0.72	3.06	0.122 4	not significant
Pure error	1.18	5	0.24			
Cor Total	253.35	19				

**Table 3 foods-11-03849-t003:** Effect of different solvents on RBO yield and oryzanol content of RBO.

Solvent Type	RBO Yield/%	Oryzanol Content/%
Mixed solvent	85.8 ± 0.0 ^a^	2.5 ± 0.0 ^a^
Isopropanol	78.6 ± 0.3 ^c^	2.5 ± 0.0 ^ab^
Cyclohexane	81.1 ± 0.2 ^b^	2.4 ± 0.0 ^b^

Note: Different lowercase letters in the columns indicate significant differences among groups (*p* < 0.05).

**Table 4 foods-11-03849-t004:** Physicochemical properties of RBO extracted using different solvents.

Parameters	Mixed Solvent	n-Hexane	Solvent No. 6
Moisture and volatiles (%)	7.2 ± 0.0 ^b^	3.1 ± 0.0 ^c^	18.9 ± 0.6 ^a^
Acid value (mg/g)	13.74 ± 0.21 ^b^	13.13 ± 4.47 ^b^	15.57 ± 0.12 ^a^
Peroxide value (mmol/kg)	8.52 ± 0.35 ^b^	8.63 ± 0.29 ^b^	9.92 ± 0.21 ^a^
Iodine value (g/100 g)	106.73 ± 0.52 ^a^	106.51 ± 0.82 ^a^	105.41 ± 0.66 ^a^
L*	25.66 ± 0.42 ^b^	28.92 ± 0.32 ^a^	28.97 ± 0.10 ^a^
a*	3.35 ± 0.10 ^a^	2.86 ± 0.02 ^b^	2.44 ± 0.04 ^c^
b*	9.85 ± 0.32 ^c^	15.51 ± 0.24 ^a^	14.49 ± 0.01 ^b^
ΔE*	67.12 ± 0.44 ^a^	64.46 ± 0.28 ^b^	64.24 ± 0.10 ^b^
Oryzanol content (%)	2.7 ± 0.1 ^a^	2.5 ± 0.1 ^b^	2.1 ± 0.1 ^c^
Total sterol content (%)	5.1 ± 0.1 ^a^	5.1 ± 0.2 ^a^	5.0 ± 0.3 ^b^

Note: Different lowercase letters in a row indicate significant differences among groups (*p* < 0.05). L*, luminosity; a*, green to red; b*, blue to yellow; ΔE*, color aberration.

**Table 5 foods-11-03849-t005:** Fatty acid composition and content of RBO extracted with different solvents.

Fatty Acids	Contents of Fatty Acid (%)		
	Mixed Solvent	n-Hexane	Solvent No. 6
Myristic acid (C14:0)	0.3 ± 0.0 ^b^	0.3 ± 0.0 ^a^	0.3 ± 0.0 ^b^
Palmitic acid (C16:0)	19.9 ± 0.3 ^a^	19.7 ± 0.2 ^a^	19.8 ± 0.0 ^a^
Palmitoleic acid (C16:1)	0.2 ± 0.0 ^a^	0.2 ± 0.0 ^b^	0.2 ± 0.0 ^b^
Margaric acid (C17:0)	0.0 ± 0.0 ^c^	0.1 ± 0.0 ^a^	0.0 ± 0.0 ^b^
Stearic acid (C18:0)	2.1 ± 0.1 ^a^	2.2 ± 0.0 ^a^	2.1 ± 0.0 ^a^
Oleic acid (C18:1)	33.7 ± 0.0 ^ab^	33.9 ± 0.3 ^a^	33.4 ± 0.1 ^b^
Linoleic acid (C18:2)	38.7 ± 0.1 ^a^	38.6 ± 0.3 ^a^	38.7 ± 0.1 ^a^
Linolenic acid (C18:3)	2.1 ± 0.1 ^a^	2.1 ± 0.1 ^a^	12.0 ± 0.0 ^a^
Arachidic acid (C20:0)	0.7 ± 0.0 ^c^	0.8 ± 0.0 ^b^	0.9 ± 0.0 ^a^
Arachidonic acid (C20:1)	0.7 ± 0.0 ^b^	0.7 ± 0.0 ^ab^	0.7 ± 0.0 ^a^
Heneicosanoic acid (C21:0)	0.1 ± 0.0 ^a^	0.1 ± 0.0 ^b^	0.1 ± 0.0 ^a^
Behenic acid (C22:0)	0.4 ± 0.0 ^b^	0.5 ± 0.0 ^a^	0.4 ± 0.0 ^b^
Erucic acid (C22:1)	0.1 ± 0.0 ^a^	ND	0.1 ± 0.0 ^a^
Lignoceric acid (C24:0)	1.1 ± 0.0 ^b^	1.0 ± 0.0 ^c^	1.3 ± 0.0 ^a^
Saturated fatty acids (SFA)	24.6 ± 0.1 ^a^	24.6 ± 0.2 ^a^	24.9 ± 0.1 ^a^
Monounsaturated fatty acids (MUFA)	34.7 ± 0.0 ^a^	34.7 ± 0.3 ^a^	34.4 ± 0.1 ^a^
Polyunsaturated fatty acids (PUFA)	40.7 ± 0.1 ^a^	40.7 ± 0.3 ^a^	40.7 ± 0.1 ^a^
Total unsaturated fatty acids (UFA)	75.4 ± 0.1 ^a^	75.4 ± 0.2 ^a^	75.1 ± 0.1 ^a^

Note: Different lowercase letters in a row indicate significant differences among groups (*p* < 0.05); ND indicates that the fatty acid was not detected.

## Data Availability

Data can be made available upon reasonable request by the corresponding author.

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
