# Peer review of "Optimization of Oil Extraction from Rice Bran with Mixed Solvent Using Response Surface Methodology"

_foods, 2022, doi:10.3390/foods11233849_

Round 1

Reviewer 1 Report

This manuscript provided a novel mixed solvent. The effects of solid-liquid ratio, extraction time, extraction temperature and oscillation speed on the extraction ratio were investigated and the optimal extraction conditions were determined. This study was helpful and important for the oil extraction in rice bran. There are some questions and suggestions as follows.

Question:

Q1: Line 73. In the manuscript, the RBO yield was chosen as the indicator, but in the following text, the RBO yield and Oryzanol content was used simultaneously.

Q2: Line 86. What is the range of rice bran particle size? It may affect the RBO yield.

Q3: Figure 1. Why did you choose these six types of solvent?

Q4: Figure 3. Are there any other influencing factors besides solid-liquid ratio, temperature, time and oscillation speed?

Q5: 3.1.1 the RBO yield and oryzanol content of No.6 solvent was the highest, why you choose the isopropanol and cyclohexane nor the No.6 solvent?

Q6: 3.1.5 there was no significant difference in oscillation speed(a), why you not choose the 90 or 120/min for lower energy consumption?

 Suggestion:

Sugg1: The format of the four formulas is not same.

Sugg2: Section 2.4. Please add references behind the determination of RBO properties.

Sugg3: Section 2.4.2. Please indicate the instrument type behind the GC-MS.

Author Response

Dear Editors and Reviewers,

Thanks for your great concern and comments on our manuscript entitled “Optimization of oil extraction from rice bran with mixed-solvent by response surface methodology” (ID: foods-2002190). These comments are valuable and helpful for revising and improving our manuscript. We have revised the manuscript carefully, and the answer to each comment has been addressed as follows.

Reviewer 1:

Comments: This manuscript provided a novel mixed solvent. The effects of solid-liquid ratio, extraction time, extraction temperature and oscillation speed on the extraction ratio were investigated and the optimal extraction conditions were determined. This study was helpful and important for the oil extraction in rice bran. There are some questions and suggestions as follows.

Detail comments:

Q1: Line 73. In the manuscript, the RBO yield was chosen as the indicator, but in the following text, the RBO yield and Oryzanol content was used simultaneously.

Response: When we used the response surface methodology-central composite design (RSM-CCD) to optimize the extraction condition, the RBO yield was only chosen as the indicator. When the effect of solvent type and mixing ratio was studied, both the RBO yield and oryzanol content were chosen as the indicators.

Q2: Line 86. What is the range of rice bran particle size? It may affect the RBO yield.

Response: The biggest size of rice bran particle was about 1400 μm and the average diameter was 213.6 μm, which was described in line 85-86.

Q3: Figure 1. Why did you choose these six types of solvent?

Response: Generally speaking, considering the RBO yield and safety of solvent, a short chain alcohol and short side chain hydrocarbon compound were finally selected. Because they have the characteristics of safe, green and high yield. All the six types of solvent meet the above characteristics. Therefore, these six types of solvent were selected finally.

Q4: Figure 3. Are there any other influencing factors besides solid-liquid ratio, temperature, time and oscillation speed?

Response: Besides the solvent type and mixing ratio of solvent, solid-liquid ratio, temperature, time and oscillation speed, there are still some other factors, such as particle size, sample moisture content, pH value and so on. However, the most important factors, which affects the extraction of rice bran oil, are solvent type, solid-liquid ratio, extraction temperature and extraction time.

Q5: 3.1.1 the RBO yield and oryzanol content of No.6 solvent was the highest, why you choose the isopropanol and cyclohexane nor the No.6 solvent?

Response: Both No.6 solvent and n-hexane are solvents which were currently used at present. The RBO yield of No.6 solvent was highest, but the difference is small, and No.6 solvent is a mixed solvent which was unhealthy and unsafety. So we chose the mixture of isopropanol and cyclohexane for the next experiment.

Q6: 3.1.5 there was no significant difference in oscillation speed(a), why you not choose the 90 or 120/min for lower energy consumption?

Response: It is true that there was no significant difference according to Pearson test, but the average value of RBO is 180 r/min, which is the highest. In order to maximize the extraction yield and reduce the impact of oscillation speed, hence the 180 r/min was chosen.

Sugg1: The format of the four formulas is not same.

Response: The four formulas were revised in the manuscript according to the reviewer’s suggestion.

Sugg2: Section 2.4. Please add references behind the determination of RBO properties.

Response: The references have been added in Section 2.4 according to your suggestion.

Sugg3: Section 2.4.2. Please indicate the instrument type behind the GC-MS.

Response: The instrument type of the GC-MS has been added in Section 2.4.2.

Reviewer 2 Report

Manuscript ID: foods-2002190

Type:  Article

Title:  Optimization of oil extraction from rice bran with mixed-solvent by response surface methodology

Authors:  Zhenhua Wang , Shuzhen Li , Min Zhang * , Huanyue Yang , Gang Li , Xin Ren , Shan Liang

The manuscript presented for revision is interesting. This work is well organized and scientifically sound.  However, I have some minor comments:  

·       page 1, line 20  "..... were close to the n-hexane and significantly lower than solvent No 6, ...". Please specify the composition of the solvent No6.

·       page 2, line 84 - How long was the milled powder from rice bran stored?

·       page3, line 97 - Was the size of the rice bran sample used for the extraction (5g) not too small, according to the authors?  Perhaps a larger sample (e.g. 10 g) would allow for greater measurement accuracy.

·       The obtained results were well analyzed, but only slightly discussed with the works published in recent years. Please complete this.

Author Response

Dear Editors and Reviewers,

Thanks for your great concern and comments on our manuscript entitled “Optimization of oil extraction from rice bran with mixed-solvent by response surface methodology” (ID: foods-2002190). These comments are valuable and helpful for revising and improving our manuscript. We have revised the manuscript carefully, and the answer to each comment has been addressed as follows.

Reviewer 2:

Comments: The manuscript presented for revision is interesting. This work is well organized and scientifically sound.  However, I have some minor comments:  

Q1: page 1, line 20 "..... were close to the n-hexane and significantly lower than solvent No 6, ...". Please specify the composition of the solvent No6.

Response: No. 6 solvent generally refers to No. 6 extraction solvent oil. In the food industry, it is mainly used for the extraction of natural spices, pigments, oils and other fat soluble substances. It is a mixture of various low-grade alkanes. The main components are as follows: The content of n-hexane, 2,4-dimethylpentane, 2,3-dimethylbutane, cyclopentane, cyclohexane, benzene and n-pentane is about 30%, 18%, 18%, 10%, 8%, 4% and 2%, respectively. There are still some low content components such as 3-methylpentane, 2,2,3-trimethylbutane and other components, which are complex in composition.

Q2: page 2, line 84 - How long was the milled powder from rice bran stored?

Response: The fresh extruded rice bran was provided by Heilongjiang Beidahuang Rice Industry Group Co., Ltd. and then quickly milled by a grinder (A11, IKA, Germany) under the protection of liquid nitrogen. The milled powder was sieved through a sieve of 20 mesh and stored at -20 °C to prevent enzymatic deterioration before the experiments. The experimental period was 1-6 months, and the milled powder from rice bran was stored for 1-6 months.

Q3: page3, line 97 - Was the size of the rice bran sample used for the extraction (5g) not too small, according to the authors?  Perhaps a larger sample (e.g. 10 g) would allow for greater measurement accuracy.

Response: According to the reference, we found that different sample weights (1 g, 3 g and 9 g, [references 1-3]) were used for extraction. Then based on the solid-liquid ratio and laboratory conditions, 5g was finally selected as the sample weight. During the experiment, high-precision electronic balance and pipette were used and we paid attention to the details of the experiment to ensure the accuracy. Your suggestions are valuable and helpful, and a larger sample (e.g. 10 g) should be selected in future for greater measurement accuracy.

[1] LI D, ZHANG J, FAIZA M, et al. The enhancement of rice bran oil quality through a novel moderate biorefining process[J]. Food science & technology, 2021,151: 112118.

[2] SAWADA K, NAKAGAMI T, RAHMANIA H, et al. Isolation and structural elucidation of unique γ-oryzanol species in rice bran oil[J]. Food chemistry, 2021,337: 127956.

[3] SOARES J F, Dal PRÁ V, de SOUZA M, et al. Extraction of rice bran oil using supercritical CO2 and compressed liquefied petroleum gas[J]. Journal of Food Engineering, 2016,170: 58-63.

Q4: The obtained results were well analyzed, but only slightly discussed with the works published in recent years. Please complete this.

Response: We have revised the discussion section of manuscript carefully according to your suggestion, and the revisions were located in section 3.1.1, 3.1.2, 3.1.3, 3.1.4, 3.1.5 (lines 303-307), section 3.2.1 (lines 337-350), and section 3.3.

Reviewer 3 Report

The current manuscript studied the optimization of oil extraction from rice bran with mixed-solvent using response surface methodology.

In my opinion, the manuscript has a well-written structure. The present manuscript can be published in Foods after minor revisions as follows:

 ·       The amount of cultivated area and global rice production should be written based on the latest FAO statistics.

·       Some words, units, and equations in the text are wrongly bold and need to be corrected.

·       Add references used in all materials and methods sections.

·       Write the features of the devices in section 2.4.2.

·       Lines 168: Change "µ g" to "µg".

·       The results of section 3.1 need to be supported by statistical results "(p<0.05) or (p>0.05)".

·       In the caption of the figures, clearly, state what the uppercase and lowercase represent.

·       Define all the parameters (X1, X2, X3, A, B, ...) in the Note of table 1.

·       In the axis title of Fig 3A, modify the Solid-liquid ratio unit from "mL/g" to "g/mL".

·       What do X1, X2, X3, and R1 represent in equation 5?

·       In some parts of the manuscript, you have used the letters A, B, C and Y to introduce the variables and response. But in the table and equations X1, X2, X3, and R are used, please make them all the same.

·       Line 290: Discuss the value of the coefficient of variation.

·       Most of the references used are old. Use more up-to-date references (2019-2022) in the manuscript.

·       The discussion section is very weak and needs to be improved.

Author Response

Dear Editors and Reviewers,

Thanks for your great concern and comments on our manuscript entitled “Optimization of oil extraction from rice bran with mixed-solvent by response surface methodology” (ID: foods-2002190). These comments are valuable and helpful for revising and improving our manuscript. We have revised the manuscript carefully, and the answer to each comment has been addressed as follows.

Reviewer 3:

Comments: The current manuscript studied the optimization of oil extraction from rice bran with mixed-solvent using response surface methodology. In my opinion, the manuscript has a well-written structure. The present manuscript can be published in Foods after minor revisions as follows:

Q1: The amount of cultivated area and global rice production should be written based on the latest FAO statistics.

Response: Thank you for your helpful suggestion. We have revised the manuscript carefully according to your suggestion. Please see line 27-29. But we don't find the latest data of 2021 or 2022, and the latest data was up to 2020, which FAO published.

Q2: Some words, units, and equations in the text are wrongly bold and need to be corrected.

Response: We are very sorry for our carelessness. The wrongly bold words, units, and equations have been revised through the whole manuscript carefully according to your suggestion.

Q3: Add references used in all materials and methods sections.

Response: Thank you for your comments. We have revised the manuscript carefully according to your suggestion. Please see line 125. Other parameters are determined according to the national standards, which have been noted in the manuscript.

Q4: Write the features of the devices in section 2.4.2.

Response: We are very sorry for our carelessness. We have revised the manuscript carefully according to your suggestion. Please see section 2.4.2. in lines 133-136.

Q5: Lines 168: Change "µ g" to "µg".

Response: We are very sorry for our carelessness. "µ g" has been changed to "µg" according to your suggestion. Please see section 2.4.4 in line 173.

Q6: The results of section 3.1 need to be supported by statistical results "(p<0.05) or (p>0.05)".

Response: We have revised the manuscript carefully according to your suggestion. Please see section 3.1.

Q7: In the caption of the figures, clearly, state what the uppercase and lowercase represent.

Response: Thank you for your helpful suggestions. We have revised the manuscript carefully according to your suggestion. Please see figure 1 in lines 200-201, figure 2 in lines 226-228 and figure 4 in lines 354-355.

Q8: Define all the parameters (X1, X2, X3, A, B, ...) in the Note of table 1.

Response: We are very sorry for our carelessness of using two type letters to represent the variables and response. We have revised the table 1 carefully according to your suggestion.

Q9: In the axis title of Fig 3A, modify the Solid-liquid ratio unit from "mL/g" to "g/mL".

Response: We are very sorry for our carelessness. We have revised the figure 3 carefully according to your suggestion.

Q10: What do X1, X2, X3, and R1 represent in equation 5?

Response: X1 represents the value of solid-liquid ratio, X2 represents extraction time, X3 represents extraction temperature, R1 represents RBO yield. We are very sorry for our carelessness of using two type letters to represent the variables and response. We have revised in table 1, equation 4 and equation 5.

Q11: In some parts of the manuscript, you have used the letters A, B, C and Y to introduce the variables and response. But in the table and equations X1, X2, X3, and R are used, please make them all the same.

Response: Thank you for your helpful suggestions. We are very sorry for our carelessness of using two type letters to represent the variables and response. We have revised the table 1, equation 4 and equation 5 according to your suggestion.

Q12: Line 290: Discuss the value of the coefficient of variation.

Response: We have revised the manuscript carefully according to your suggestion. Please see the lines 337-350.

Q13: Most of the references used are old. Use more up-to-date references (2019-2022) in the manuscript.

Response: Thank you for your helpful suggestion. We have revised the discussion and the references have been updated according to your suggestion.

Q14: The discussion section is very weak and needs to be improved.

Response: Thank you for your helpful suggestion. The discussions have been enriched according to your suggestion in section 3.1.1, 3.1.2, 3.1.3, 3.1.4, 3.1.5 (lines 303-307), section 3.2.1 (lines 337-350) and section 3.3.

Round 2

Reviewer 2 Report

I sincerely thank the Authors for responding to my comments. In my opinion, the explanations are sufficient and, taking these responses into account, the article may be adopted for further processing.

Author Response

Dear Reviewers, 

Thank you for your great comments and they are valuable and helpful for revising and improving our manuscript. 

Reviewer 3 Report

The revised manuscript is suitable for publication in Foods.

Author Response

(The authors gave the same response as above.)
